# Low Plasma Choline, High Trimethylamine Oxide, and Altered Phosphatidylcholine Subspecies Are Prevalent in Cystic Fibrosis Patients with Pancreatic Insufficiency

**DOI:** 10.3390/nu17050868

**Published:** 2025-02-28

**Authors:** Wolfgang Bernhard, Anna Shunova, Julia Boriga, Ute Graepler-Mainka, Johannes Hilberath

**Affiliations:** 1Department of Neonatology, Children’s Hospital, University Clinic, 72076 Tübingen, Germany; anna.shunova@med.uni-tuebingen.de; 2General Pediatrics, Hematology & Oncology, Children’s Hospital, University Clinic, 72076 Tübingen, Germany; julia.boriga@web.de (J.B.); ute.graepler-mainka@med.uni-tuebingen.de (U.G.-M.); 3Department of Pediatric Gastroenterology and Hepatology, Children’s Hospital, University Clinic, 72076 Tübingen, Germany; johannes.hilberath@med.uni-tuebingen.de

**Keywords:** betaine, choline deficiency, cystic fibrosis, exocrine pancreas insufficiency, hepatosteatosis, PEMT, phosphatidylcholine, SIBO, TMAO

## Abstract

Background: Exocrine pancreatic insufficiency in cystic fibrosis (CF) increases fecal choline losses, but the postnatal course of plasma choline and its metabolites in these patients is unknown. While choline homeostasis is crucial for cellular, bile, and lipoprotein metabolism, via phosphatidylcholine (PC) and via betaine as a methyl donor, choline deficiency is associated with impaired lung and liver function, including hepatic steatosis. Objective: The goal of our study was to evaluate the plasma levels of choline, betaine, trimethylamine oxide (TMAO), PC, and PC subclasses in CF patients from infancy to adulthood and compare those with exocrine pancreatic insufficiency (EPI) to those with pancreatic sufficiency (EPS). Methods: Retrospective analysis of target parameters in plasma samples (July 2015–November 2023) of CF patients (0.64–24.6 years) with tandem mass spectrometry. Results: A total of 477 samples from 162 CF patients were analyzed. In CF patients with EPI (N = 148), plasma choline and betaine concentrations were lower and decreased with age compared to EPS patients showing normal values. TMAO concentrations, indicating intestinal choline degradation by bacterial colonization, were frequently elevated in EPI from infancy onwards, and inversely related to plasma choline and betaine levels. PC-containing linoleic acid levels were lower in EPI, but arachidonic and docosahexaenoic acid content was similar in both patient groups. Conclusion: CF patients with EPI are at risk of choline and betaine deficiency compared to exocrine pancreas-sufficient CF patients. Elevated TMAO concentrations in EPI patients indicate increased bacterial colonization leading to choline degradation before absorption. These findings indicate that laboratory testing of choline, betaine, and TMAO as well as clinical trials on choline supplementation are warranted in CF patients.

## 1. Introduction

Cystic fibrosis (CF) is a recessive autosomal disease with an incidence of 1:3300 to 1:4800 in the Caucasian population. It is caused by mutations in the Cystic Fibrosis Transmembrane Conductance Regulator (CFTR) gene, mostly characterized by the homozygous or compound heterozygous F508del genotype, comprising about 80% of patients in the Caucasian population. There are approximately 200 new CF cases per year in Germany, with a prevalence of 6000–7000 patients. Prevalence is 30,000–35,000 in all Europe as well as in North America, but lower in non-Caucasians [1]. Organ manifestations, such as impaired structure and function of the exocrine pancreas, intestine, and lungs begin to develop in utero [2]. Postnatally, development is impaired despite state-of-the-art therapies, including digestive enzyme substitution in exocrine pancreatic-insufficient patients. Lung disease is the most frequent, and CF-associated liver disease (CFALD) the second-most cause of early death [3].

Of all CF patients, 85–87% develop exocrine pancreas insufficiency (EPI), mostly in utero or in early infancy, and there is a strong link between exocrine pancreas and lung function [4]. Notably, CF patients with EPI show increased fecal loss of the essential nutrient choline, and frequently low plasma concentrations of choline and PC [5,6]. Hence, choline deficiency potentially impacts ~5000–6000 CF patients in Germany and ~25,000–30,000 in Europe as well as in the USA. Pancreatic enzyme substitution, adequate nutrition, and antibiotic treatment have increased the median life expectancy of these patients to about 50 years [7]. Although pulmonary failure is the most frequent cause of death, CFALD, a synopsis of alterations from hepatic steatosis to biliary cirrhosis with potentially fatal consequences, is the second-most frequent cause of death and is associated with pancreatic insufficiency [8]. As hepatic steatosis and/or cholestasis are known consequences of choline deficiency, particularly in choline-free parenteral nutrition (PNALD = parenteral nutrition associated liver disease) [9], chronic choline deficiency may contribute to CFALD as well. However, only limited data on the choline status of CF patients in relation to age, and no systematic investigation comparing patients with and without exocrine pancreas insufficiency, are available.

CFTR functions as a transporter of chloride (Cl^−^), hydrogen carbonate (HCO_3_^−^), sphingosine-1-phosphate (S1P), and γ-glutamyl-cysteinyl-glycine (reduced glutathione, GSH). These are all altered in CF and related to choline metabolism: via HCO_3_^−^, requiring CFTR-mediated Cl^−^ transport for the Cl^−^/HCO_3_^−^ antiporter, CFTR is crucial for physiological pH regulation in the small intestine, mucin solubilization, and the optimal function of pancreatic phospholipase A_2_ IB (sPLA_2_IB). This enzyme is essential for the enterohepatic cycle of bile PC and, therefore, EPI impacts choline homeostasis [5,10]. Moreover, via S1P and other components, CFTR contributes to sphingolipid metabolism and, therefore, to choline-containing sphingomyelin (SPH) and apoptosis. Finally, homeostasis of the antioxidant GSH, which is also decreased in CF, is linked to choline via betaine and its downstream metabolites as methyl donors for the methylation of homocysteine to methionine. Its activation to S-adenosylmethionine (SAM) is required for stimulating the hepatic trans-sulfuration pathway resulting in cysteine and GSH formation [11].

Quantitatively, choline as an essential nutrient is predominantly required for parenchymal homeostasis in the form of membrane phospholipids, such as PC and SPH, and for the formation of betaine as a methyl donor. SPH is high in cell membranes and high-density lipoproteins, whereas PC dominates in bile and very low-density lipoproteins (VLDL). Hepatic PC turnover amounts to ~50% of its PC pool per day via bile secretion, along with the PC moiety of VLDL for hepatic triglyceride export [12]. The irreversible synthesis and turnover of betaine as a methyl donor account for ~40% of ingested choline which significantly contributes to choline requirement [13].

Key to choline deficiency in CF patients is the loss of bile PC via feces due to EPI [5,14]. Plasma concentrations of choline and betaine are low in these CF patients compared to fasting controls (6.4 ± 0.3 vs. 7.8 ± 0.3 µmol/L and 20.9 ± 2.1 vs. 30.3 ± 4.2 µmol/L, respectively) [6], and may be extremely low (~2 µmol/L) in some individuals with severe hepatic steatosis [15]. Additionally, the sticky intestinal mucus and impaired motility makes CF patients susceptible to small intestinal bacterial colonization and overgrowth (SIBO) [15]. Hence, choline assimilation could additionally be impaired by bacterial degradation to trimethylamine (TMA) prior to absorption, followed by the oxidation of TMA to TMAO in the liver. The contribution of this non-genetic factor is suggested by increased plasma concentrations of TMAO in adult CF patients with EPI [15,16]; however, there is a lack of systematic evaluation of choline deficiency, SIBO, and their relation in CF disease. A pilot interventional study conducted on adult CF patients confirmed the relationship between choline homeostasis, liver steatosis, and pulmonary function; the observed clinical improvements were most significant in patients whose organs were more severely affected [16].

From a pathophysiological perspective, decreased small intestinal pH and impaired function of the highly pH-sensitive pancreatic sPLA2IB result in a lack of PC cleavage to lyso-PC, thereby blunting choline re-uptake via the enterohepatic cycle in the terminal ileum, resulting in fecal choline loss. About 50% of the hepatic PC pool is secreted daily into the duodenum, accounting for ~11–12 of 23 g PC in a 1500 g adult liver [12], equaling ~1.6 g choline, i.e., 3–4-fold of the adequate intake (AI) of adults (400–550 mg/d) [17,18]. Finally, as the AI of choline is mostly not achieved by the population, its enterohepatic turnover may exceed the value of 3–4-fold of its daily intake [19]. Hence, dysfunction of the exocrine pancreas and the enterohepatic cycle together with bacterial choline degradation (see above) may result in choline deficiency in CF.

In addition, PC is essential for hepatic triglyceride secretion and the transport of long-chain poly-unsaturated fatty acids (LC-PUFA) via VLDL, comprising 20% PC [20]. Hence, via the PC moiety of VLDL, choline, and LC-PUFA metabolism of the liver are linked to that in other organs including the lungs [21].

Endogenous hepatic PC synthesis via the phosphatidylethanolamine N-methyltransferase (PEMT) pathway (requiring SAM) is not sufficient to meet choline requirements [18,19]. Moreover, decreased PEMT activity is characteristic of CF patients [6]. Additionally, frequent single-nucleotide polymorphisms (SNPs) in the PEMT gene further increase exogenous choline requirements and are associated with (non-alcoholic) fatty liver disease [22].

However, no systematic overview of choline homeostasis, including that of its metabolites, exists to date for CF patients of different ages. Although an association between exocrine pancreatic function and choline status has been suggested [5,14], it has never been evaluated on a routine basis in patients of different ages nor in comparison with pancreas-sufficient CF patients. However, this is important, as pancreatic insufficiency—and poor sPLA2IB function—develops very early, and correcting for choline deficiency may be important for the development of these patients [15,16]. Therefore, we retrospectively investigated the plasma concentrations of choline and its metabolites in relation to exocrine pancreatic function in CF patients from infancy to adulthood.

## 2. Materials and Methods

### 2.1. Study Population

Retrospective analysis of patient data was approved by the Institutional Review Board (121/2020BO2). As a pseudonymized, retrospective study, written consent of patients was not required. Routine assessment of choline and fatty acid status in patients with CF was introduced in our outpatient clinic in 2015. Exclusion criteria for participation in this study were absence of genetic verification of CF, acute liver damage with hepatocyte decay (prior to liver transplant), verified intake of choline as a prescribed choline supplement or as part of medication, age 25 year or beyond, and missing choline data (Table 1, Figure 1).

### 2.2. Exclusion of Additional Choline Intake

Additional choline intake via medication (3000 mg choline chloride in adults) or via dietary supplements (such as Fortimel^®^ [Danone Deutschland GmbH, Frankfurt, Germany], Combiotic^®^ [HiPP, Pfaffenhofen, Germany], or Energea P Kid [metaX, Friedberg, Germany], only the latter being free of choline) was retrieved from the patient files. Plasma choline concentrations showed a weak correlation with the additional intake of choline supplements or medication (0–220 mg/d, r = 0.1470, *p* = 0.0007) (Appendix A). Therefore, although at the given choline amounts via formula or supplements plasma effects were low compared to interventions with 2200 mg/d [15,16], samples during any kind of choline supplementation were excluded. For further patient and inclusion characteristics, see the flow chart (Figure 1) and Table 1.

### 2.3. Sample Parameters and Numbers

Of the total 611, 477 EDTA plasma samples (July 2015 to November 2023) from 162 CF patients aged 0 to <25 years, without (N = 14) or with (N = 148) exocrine pancreatic insufficiency, were included in this study (see flow sheet Figure 1). Patients were routinely screened for plasma concentrations of choline, betaine, TMAO, PC, lyso-PC, SPH, and ceramides. PC was further differentiated into subclasses as defined by their fatty acid content, that is, comprising either two saturated fatty acids (disat.-PC) or an oleic acid (OA) (C18:1-PC), linoleic acid (LA) (C18:2-PC), arachidonic acid (ARA) (C20:4-PC), eicosapentaenoic acid (EPA) (C20:5-PC), or docosahexaenoic acid (DHA) residue (C22:6-PC), together with that of a saturated fatty acid (mainly palmitic or stearic acid) [16].

### 2.4. Plasma Collection

EDTA blood (1–2.7 mL) was harvested from venous puncture after >4 h fasting, kept on crash ice, and centrifuged within 1 h at 1000× *g* for 10 min at room temperature. Plasma supernatant was immediately aspirated and aliquoted in 100 µL samples, frozen at −20 °C, and transferred to a −80 °C freezer within 5 days until analysis.

### 2.5. Mass Spectrometry

Plasma samples were processed using established standard procedures as previously described [23,24]. In brief, plasma was spiked with internal standard (D_4_-choline chloride) and extracted with chloroform/methanol according to Bligh and Dyer [25]. The upper water/methanol phase, containing the water-soluble metabolites (choline, betaine, TMAO, and others), was separated from the lipid-containing chloroform phase. Diarachidoyl-PC (PC20:0/20:0) as a mass spectrometry standard was added after phase separation to an aliquot of the chloroform extract, as Bligh and Dyer extraction is quantitative [25], to obtain sample aliquots without additional arachidic acid for potential gas chromatography.

Equipment for analysis comprised a TSQ Quantum Discovery Ultra tandem mass spectrometer, Finnigan Surveyor Autosampler Plus, and Finnigan Surveyor MS Pump Plus (Thermo Fisher Scientific, Dreieich, Germany). Choline, D_4_-choline, betaine, and TMAO were separated on a ZORBAX HILIC Plus Narrow Bore RR column (2.1 × 100 mm inner diameter; 3.5 µm particle size) (Agilent Technologies, Waldbronn, Germany) at 35 °C. Elution was performed at 0.4 mL/min with solvent A (acetonitril/water:50 mM ammonium formate; 74:21:5; *v*/*v*) and B (water/50 mM ammonium formate, 95:5; *v*/*v*): gradient conditions were 100% A (0–0.1 min) → 32% B (0.1–5.25 min) → 100% A (5.25–5.4 min) → 100% A (5.4–12 min), and components were analyzed at positive ionization. PC, lyso-PC, and SPH were separated isocratically on a Polaris Si-A (2 × 150 mm i.d.; 2 µm; Agilent Technologies) with chloroform/methanol/300 mM ammonium acetate (60:38:2%, *v*/*v*) as the mobile phase. Phosphorylcholine (mass/charge [*m*/*z*] = +184) served as the diagnostic fragment.

### 2.6. Clinical Parameters

Clinical data were extracted from the hospital’s software SAP (Systemanalyse Programmentwicklung; Version 2020 and 2021, SAP SE & Co., Ltd. KG, Walldorf, Germany) and the documentation system of the Mucoviszidosis outpatient clinic ARDIS2 (Arthritis und Rheumatologie Dokumentations- und Informationssystem 2; Version 1.52.2, axaris-software & systeme GmbH, Dornstadt, Deutschland, 2009–2022). Lung function parameters measured were forced volume vital capacity (FVC), percentage of predicted forced expiratory volume in 1 s relative to FVC (ppFEV1), and forced expiratory flow at 25% and at 25–75% of the pulmonary volume (FEF25, FEF 25–75).

### 2.7. Statistics

Data management was performed using Microsoft Office Excel version 2021. To eliminate unbalanced documentation of individual patients with multiple determinations (2–9) and to reduce errors due to single extremely low or high values, the medians of an individual’s data were formed, if more than one sample was available, resulting in 162 data points, of which 14 were from pancreas-sufficient and 148 from pancreas-insufficient patients. All analytical and clinical data are expressed as medians and interquartile ranges. Significance values were determined by non-parametric testing for group comparison (Mann–Whitney U-statistic) and non-parametric correlation (Spearman’s rank correlation coefficient), using GraphPad InStat, version 3.10 (STATCON GmbH, Witzenhausen, Germany). *p*-values < 0.05 were considered significant.

## 3. Results

Biometric, routine laboratory, and lung function data are shown in Table 2. The study groups showed similar median ages and age ranges, regardless of exocrine pancreatic (in-)sufficiency (EPI or EPS), with no significant statistical difference (*p* > 0.05). A significant difference was observed in the genotype distribution, with EPI patients predominantly exhibiting F508del homozygous or compound heterozygous, whereas EPS patients were largely characterized by other CFTR genetic variants (Table 2). Most clinical parameters showed no differences between CF patients with and without exocrine pancreatic-insufficiency, except the earlier diagnosis (*p* = 0.0009), higher alanine aminotransferase (ALA) (*p* = 0.0021), and decreased FEF25-75 (*p* = 0.045) in EPI patients. The lower median FEF25 did not reach significance and ppFEV1 as well as PEF were identical in EPI compared to EPS patients. C-reactive protein (CRP) and erythrocyte sedimentation rate (ESR) were only determined in cases of suspected inflammatory exacerbations, ad showed no difference between EPS and EPI patients (Table 2). To determine any impact of inflammation on choline parameters, we correlated individual values with C-reactive protein (CRP; N = 274) and erythrocyte sedimentation rate (ESR: N = 227) levels. There was no significant correlation (*p* > 0.05; see Appendix A), but a trend between increased ESR and low choline and betaine (Appendix A).

### 3.1. Parameters of Choline Homeostasis in the Whole Study Group

Plasma concentrations of both choline and betaine were significantly lower in the EPI group (Table 3A). By contrast, concentrations of TMAO, showing a large variability, were significantly higher in these patients, whereas in EPS CF patients TMAO was <3 µmol/L throughout. Phosphatidylcholine (PC) concentrations, other components containing a choline headgroup (lyso-PC, sphingomyelin [SPH]) and ceramides did not differ between the patient groups (Table 3B). However, the pattern of PC subclasses was different (Table 3C): in EPI patients, PC species containing two saturated fatty acid residues (disaturated PC), an oleic acid (OA/C18:1) or an eicosapentaenoic acid (EPA/C20:5) residue were slightly increased, which was at the expense of PCs comprising a linoleic acid (LA/C18:2) residue or other (minor) PC compounds (Table 3B). Notably, there were no differences in PC subclasses comprising an arachidonic (ARA/C20:4) or docosahexaenoic acid (DHA/C22:6) residue.

### 3.2. Age-Related Changes of Water-Soluble Choline Compounds in Plasma

Figure 2A–C shows the median individual concentrations of plasma choline, betaine, and TMAO in relation to age (0.64–24.6 years) in CF patients with and without exocrine pancreas insufficiency. There was no age-dependent change in plasma choline or betaine levels in pancreas-sufficient patients, whereas in CF patients with exocrine pancreatic insufficiency, both choline and betaine (Figure 2A,B) as well as their sum (Appendix A) decreased with age. TMAO remained low in exocrine pancreas sufficiency throughout (<3 µmol/L), but in insufficient patients the median was higher (Table 3A), but with a large range of values, from near zero to more than 10 µmol/L. There was no correlation with age (*p* = 0.1993), and values were significantly increased (7.8 and 12.7 µmol/L) even in two infants (0.64 and 0.96 years, respectively). There was a direct correlation between plasma concentrations of betaine and choline but an inverse correlation between plasma TMAO and choline concentrations. However, while this inverse correlation was significant (*p* < 0.001), it was only weak (ρ = −0.2666) (Figure 2D). This correlation similarly applied to betaine alone and the sum of choline + betaine vs. TMAO (Appendix A).

### 3.3. Age-Related Changes of PC and Its Sub-Groups

Figure 3A shows that PC concentrations were identical in CF individuals with or without exocrine pancreatic insufficiency and did not change with age. However, there were distinct differences in the molecular composition of PC, that is, its fatty acid pattern: whereas in EPI, the fraction of disaturated PC increased with age (Figure 3B), that of PC containing an oleic acid (OA) residue (C18:1-PC) remained constant (Figure 3C). By contrast, PC containing a linoleic acid (LA, C18:2*n*-6) residue (LA-PC) decreased with age in CF patients with EPI only (Figure 3D). However, although arachidonic acid (ARA, C20:4*n*-6) is synthesized from LA, decreased LA-PC had no impact on ARA-PC.

Among the PC species containing a long-chain poly-unsaturated (LC-PUFA) fatty acid residue, those comprising an ARA, eicosapentaenoic (EPA, C20:5*n*-3), or docosahexaenoic acid (DHA, C22:6*n*-3) residue (ARA-PC, EPA-PC, and DHA-PC, respectively) are shown in Figure 4A–C. There was no difference between CF patient groups, or change with age, in ARA-PC and DHA-PC. However, with age, EPA-PC increased, which was statistically significant only in patients with pancreatic insufficiency (Figure 4B). There was no difference in the ratio between PC comprising the omega-6 fatty acid ARA versus PC containing omega-3 fatty acids (EPA, DHA) throughout (Figure 4D).

## 4. Discussion

This study addresses the plasma concentrations of choline and its derivatives, as indicators of choline deficiency, in CF patients with (EPI) (N = 148) and without (EPS) (N = 14) exocrine pancreatic insufficiency. This imbalance between patient numbers is caused by the fact that EPI patients comprise more than 85%. The rationale for our study is that low plasma choline in CF patients and the relation between choline deficiency and hepatosteatosis are well described [5,6,9], but so far not systematically investigated in CF across different ages. Imbalances of other metabolites related to choline and betaine homeostasis, however, such as increased homocysteine and decreased methionine, SAM, and glutathione [26,27,28], are well described in patients with EPI, which constitutes the majority of CF patients [1]. Choline deficiency was attributed to chronic fecal choline losses, resulting from these patients’ inefficient enterohepatic circulation of PC released into the duodenum via biliary secretion [11,12]. A clinical impact of choline deficiency and altered PC/lipoprotein metabolism in CF has been suggested so far in only two pilot studies and one case report [14,15,16]. A dose of 2200 mg rather than 550 mg per day for healthy male adults (Institute of Medicine/National Academy of Medicine) was effective in improving lung function and resolving severe hepatic steatosis [15,16]. Importantly, choline was most effective on lung function improvement if ppFEV1 values were low [16], whereas actually lung function of CF patients is mostly very good, particularly owing to recent advances in modulator treatment (see Table 2 and [29]). Nevertheless, during choline deficiency, choline drains from the lungs to the liver via high-density lipoproteins, which in the long run may affect lung parenchyma homeostasis and repair via impaired sphingolipid homeostasis [11,21]. Interestingly, within our cohort of CF patients, those with pancreatic insufficiency demonstrated significantly lower FEF25 measurements (see Table 2). However, we found no significant correlation between choline parameters and CRP, and only a trend for increased ESR vs. low choline and betaine. It is unclear whether increased ESR is associated with low choline via increased oxidative stress causing inflammation [30].

### 4.1. Plasma Choline and Betaine Levels in Relation to Exocrine Pancreas Function

In our study, we showed that only CF patients with EPI had decreased plasma levels of choline, who mostly were F508del compound heterozygous (53%) or homozygous (36%) (Table 2). Notably, all patients with EPI in our group received pancreatic enzyme replacement therapy (PERT). This observation highlights that PERT alone may not be effective in preventing choline deficiency. Of the total, 92% of our studied patients suffered from EPI, so that only 14 pancreas sufficient CF patients could be investigated. While this impairs the statistical significance for some parameters, such as the increase in EPA with age, which was significant only for the 148 EPI patients, the data clearly show that low plasma levels of choline and betaine are characteristic for the EPI patient group. Moreover, the correlation between choline and betaine levels demonstrates that low choline is an indicator of low availability of betaine as an essential methyl donor for homocysteine degradation and for the generation of SAM as the most important methyl group donor involved in glutathione (GSH) and creatine synthesis [14]. In this context, it was shown that choline supplementation increased creatine concentrations in muscle tissue [16].

### 4.2. The Impact of Small Intestinal Bacterial Colonization and PEMT Genetics

Moreover, variable but frequently high plasma levels of TMAO in EPI CF patients from infancy onwards, and its inverse correlation with choline and betaine levels, suggest that small intestinal bacterial colonization or overgrowth (SIBO) may affect choline bioavailability at all ages. However, we did not directly measure SIBO via culture-based quantification of the bacterial flora present in small bowel aspirates. Hence, high TMAO may not only be derived from choline degradation in the small intestine, but also from compounds containing choline that enter the colon and are degraded there to TMA and absorbed [5]. In addition to fecal and bacterial choline depletion, impaired endogenous (indirect) choline synthesis by methylation of phosphatidylethanolamine (PE), which requires SAM and PE-N-methyltransferase (PEMT), may contribute to choline deficiency in EPI of CF patients: while the PEMT pathway is generally decreased in these patients [6], frequent single nucleotide polymorphisms (SNPs) of the PEMT gene (such as rs12325817; 25–44% of the population) [22] may further impact the severity of the clinical consequences of choline deficiency. This was shown in a CF patient with very low plasma choline (2 µmol/L) and progressive liver steatosis from preschool age onwards, which was effectively treated with choline chloride. This patient not only suffered from SIBO but was also homozygous for the frequent PEMT SNP rs12325817 [15].

The impact of choline deficiency on liver disease has been supported by other studies, demonstrating that low plasma choline concentrations are associated with liver disease, particularly hepatic steatosis, in parenterally fed patients, and is resolved by parenteral or enteral choline supplementation [31,32]. Despite no routine liver function test in our study cohort, CF-associated liver disease cannot be excluded. There are varying recommendations for screening for liver involvement in CF including laboratory-based scores, histopathology and imaging [33]. However, in our retrospective study, we could not systematically assess and evaluate for the presence of liver disease such as hepatic steatosis. Therefore, we were not able to compare EPI and EPS patients with regard to clinical signs of CFALD. Future studies are needed to assess for both the presence of choline deficiency in CFALD with and without pancreatic insufficiency, and for the effect of choline supplementation.

### 4.3. Effects of Age on Plasma Choline and TMAO Levels

Reference values in relation to age show that plasma choline levels are high after birth, but postnatally rapidly decrease from 48 ± 15 µmol/L at birth to 12.8 ± 2.0 µmol/L at 3 years and 8.4 ± 3.1 µmol/L in adults [34]. These values were achieved by EPS, but not by EPI CF patients (Table 3). Betaine levels were similarly decreased in patients with EPI, whereas sufficient patients were similar to healthy controls [6]. The increased levels of TMAO demonstrate that the homeostasis of choline and betaine in EPI patients is linked to an overall altered microbial intestinal colonization: TMAO is an indicator of non-quantitative choline absorption, due to the presence of increased numbers of choline-cleaving bacteria in the small intestine, i.e., small intestinal bacterial overgrowth (SIBO). Here, bacteria cleave choline (and related compounds with trifold methylated ammonium groups, such as betaine and carnitine) prior to absorption. They release trimethylamine, which is subsequently absorbed and oxidized to TMAO in the liver, indicating increased intestinal choline degradation. We found no age-related differences in TMAO levels, which is consistent with the early postnatal development of FMO3 expression [35,36]. However, from our data, it remains unclear whether differences in intestinal maturation or the patients’ microbiota contribute to TMAO levels in plasma.

Expression of the responsible flavin-containing trimethylamine monooxygenase (FMO3; EC 1.14.13.148) starts postnatally [35], so that TMAO formation is nearly absent in preterm infants [37]. However, it is unclear whether the increase in postnatal FMO3 expression in CF patients is as rapid as in healthy infants and identical in all patients. Therefore, it is unclear whether low TMAO levels represent the absence of SIBO in CF patients, and to which degree differences in the individual microbiota contribute to TMAO levels. Nevertheless, the inverse correlations between TMAO and choline and betaine concentrations suggest that impaired choline bioavailability due to intestinal degradation starts at infant age and persists until adulthood (see results and [37,38]). Moreover, it is unclear whether early SIBO and TMAO formation are associated with the development of CF-associated liver disease (CFALD) development.

### 4.4. Plasma Phospholipids and Long-Chain Poly-Unsaturated Fatty Acids (LC-PUFA) in CF

While choline and betaine values were decreased and those of TMAO were increased in CF patients with EPI, plasma concentrations of phospholipids (PL), that is PC, lyso-PC, and SPH, were similar in both CF groups (1.2 to 1.5 mmol/L). Hence, the characteristically lower PL values of CF patients compared to non-CF infants and healthy adults [6,15,16] applies to all CF patients. Previous research on patients receiving parenteral nutrition demonstrated that choline deficiency as indicated by low free plasma choline concentrations can occur independently of normal or elevated plasma phospholipid (PL) levels. Notably, symptoms of choline deficiency, such as hepatic steatosis, were not alleviated by high PC concentrations in plasma, but only by free choline [32,34]. In line with this, animal experiments have shown that the delivery of free plasma choline from PC stores is low [39].

Nevertheless, the fatty acid pattern of PC was different in CF patients with EPI compared to those with EPS. Notably, there was no difference in PC containing ARA or DHA residues. In contrast, LA-PC was significantly decreased in EPI patients (see Figure 2 and Figure 3). The fraction of LA-PC decreased with age in pancreas-insufficient CF patients, whereas EPA increased. However, it is unclear whether this is due to dietary differences in CF patient groups, and it apparently has no impact on the homeostasis of ARA that is synthesized from LA. ARA and DHA remained constant and were not related to exocrine pancreas function. As LA is partly and EPA, ARA, and DHA are primarily transported via PC in plasma [40], these data show that DHA, ARA, and the omega-6-to-omega-3 ratio do not change with age in EPI patients, although EPA increases whereas LA decreases with age. These clinical data are contradictory to CF animal experiments, describing an increase in the ARA/DHA and a decrease of the EPA/LA ratio [41]. In contrast, our data confirm that from infancy to adulthood, irrespective of exocrine pancreas function, there are no general alterations in ARA and DHA compared to non-CF patients [6,16].

### 4.5. Perspectives of Clinical Choline Analysis and Application

Only few data on the benefit of choline administration in PNALD and CFALD are available [9,15,16,31,32], and the analysis of choline, together with its related metabolites including TMAO is not established in clinical routine nor is choline mentioned in actual guidelines [42]. While quantification requires only small amounts of plasma (<50 µL) and is very specific, it requires liquid chromatography and tandem mass spectrometry (LC-MS/MS) [23,24]. However, such analysis may be important for the management of CF patients with CFALD.

Increased fecal choline loss, low plasma concentrations of choline and betaine and the association between choline and steatosis/CFALD suggest that daily choline requirements of CF patients with EPI are increased over AI values [17,18] at all ages. Randomized prospective clinical trials are warranted to assess the clinical efficacy of choline supplementation for CFALD prevention and therapy on a routine basis.

Administration as choline chloride (3 g = 2.2 g choline) to adult CF patients in three dosages together with liquid and a meal showed no adverse effects [9,14,15,16]. For practical application, it must be considered that plasma turnover of choline is very rapid, and choline salts are osmotic; thus, distribution over meals and intake with liquid (or as a continuous diluted infusion) is important. To date, water-soluble physiological choline compounds like glycerophosphocholine and phosphocholine are not in use or available, whereas choline bitartrate may be disadvantageous over choline chloride due to increased intestinal TMAO formation [43].

### 4.6. Limitations

A limitation of this study is its retrospective nature, with several parameters missing, like standardized food protocols to assess the intake of choline and other food items like LC-PUFA. Another limitation is that the group of CF patients with EPS is only 14 patients. However, a matched-pair analysis would have resulted in only 14 rather than 148 EPI cases, the majority of CF patients. Further limitations—and perspectives for future studies—are that no routine diagnostic data of small intestinal colonization and of liver fat using magnetic resonance spectroscopy or elastography were available for this study.

## 5. Conclusions

Low plasma concentrations of choline and betaine, together with high TMAO, are characteristic for CF patients with EPI despite PERT. In both EPI and EPS patients, plasma phospholipid concentrations were lower than in healthy controls. There are age-dependent changes in the fatty acid profile of plasma PC, primarily concerning LA and EPA, but not ARA and DHA. There was no age-dependent alteration in the ARA/EPA + DHA ratio.

Low plasma concentrations of choline and betaine, together with high TMAO levels, in CF patients with EPI, and clinical data on the successful treatment of CF-related and unrelated steatosis by choline supplementation suggest increased choline requirements in CF patients with EPI, the implementation of choline/betaine/TMAO determination into clinical routine, and randomized clinical trials on choline supplementation in patients with CF-related liver disease.

## Figures and Tables

**Figure 1 nutrients-17-00868-f001:**
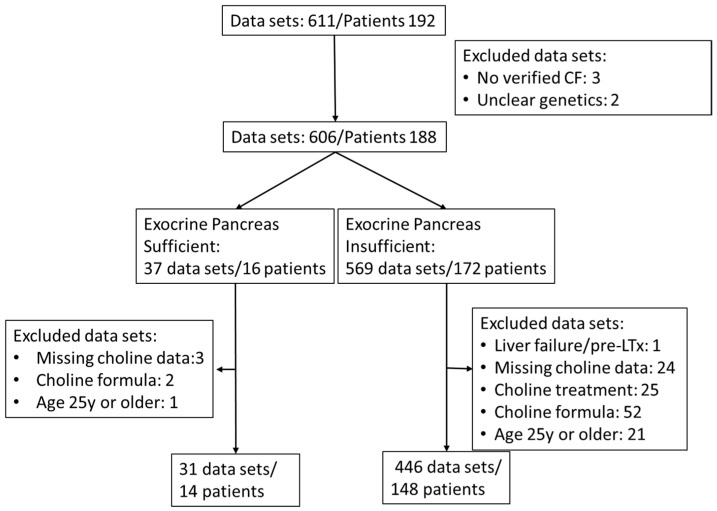
Flow chart of sample and patient inclusion.

**Figure 2 nutrients-17-00868-f002:**
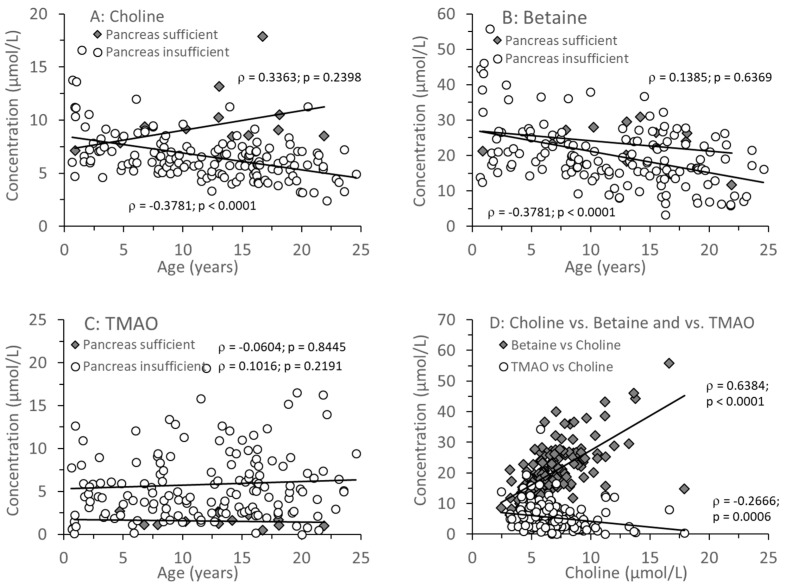
Plasma concentrations of choline and its metabolites in CF patients with (EPI) and without (EPS) exocrine pancreatic insufficiency. Choline (**A**), betaine (**B**), and trimethylamine oxide (TMAO) (**C**) are shown in relation to age. (**D**) shows the direct correlation between choline and betaine as well as the inverse relation between choline and TMAO in plasma. Data are median values of individual CF patients with (N = 148) and without (N = 14) exocrine pancreas insufficiency. Abbreviations: ρ, Spearman’s correlation coefficient; *p*, significance level.

**Figure 3 nutrients-17-00868-f003:**
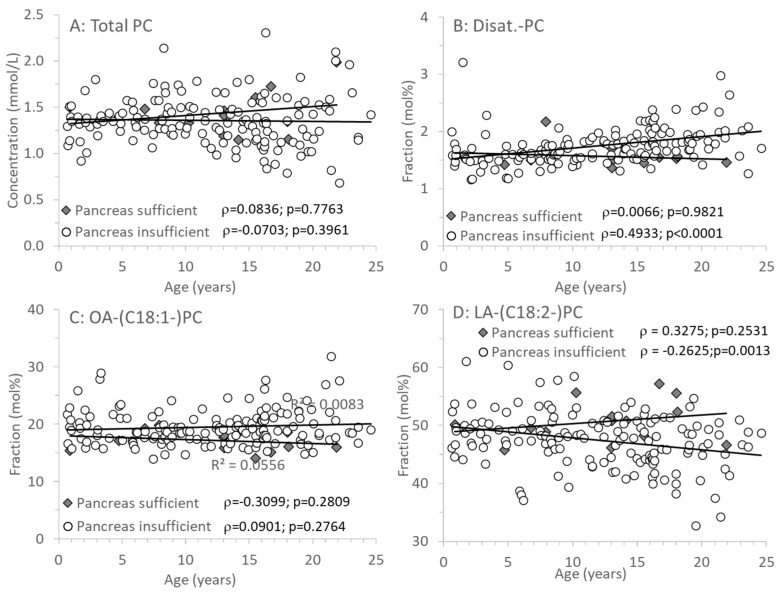
Plasma phosphatidylcholine (PC) of CF patients with (EPI) and without (EPS) exocrine pancreatic insufficiency. (**A**) shows concentrations of total PC, whereas (**B**–**D**) show its fractions of PC containing 2 saturated (Disat.-PC), an oleic acid (OA-PC) or a linoleic acid (LA-PC) residue. Data are median values of individual CF patients with (N = 148) and without (N = 14) exocrine pancreas insufficiency. Abbreviations: ρ, Spearman correlation coefficient; *p*, significance level.

**Figure 4 nutrients-17-00868-f004:**
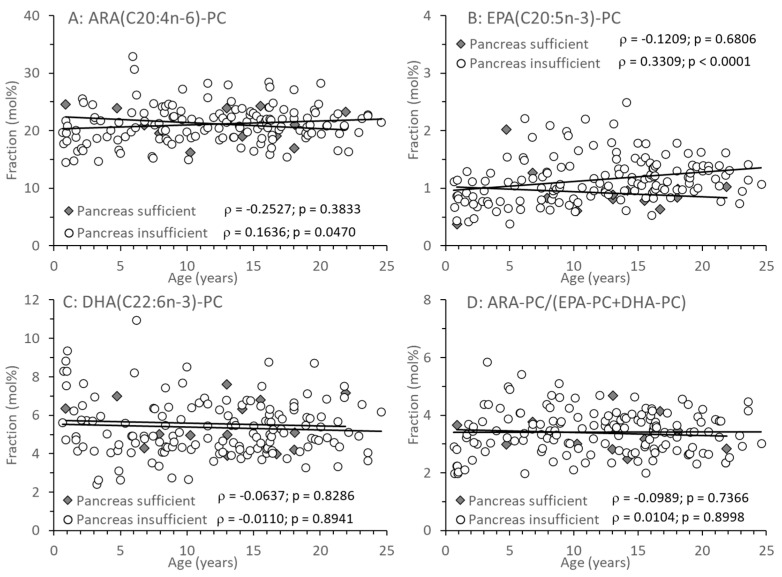
Fractions of total plasma phosphatidylcholine (PC) containing an arachidonic (ARA-PC) (**A**), eicosapentaenoic acid (EPA-PC) (**B**) or docosahexaenoic (DHA-PC) (**C**) acid residue. (**D**) shows the ratio of *n*-6 LC-PUFA vs. *n*-3 LC-PUFA-PC (ARA-PCvs.EPA-PC + DHA-PC). Data are median values of individual CF patients with (N = 148) and without (N = 14) exocrine pancreas insufficiency. Abbreviations: ρ—Spearman correlation coefficient; *p*—significance level.

**Table 1 nutrients-17-00868-t001:** Inclusion and exclusion parameters of study patients.

Inclusion parameters	Genetically verified cystic fibrosis with or without exocrine pancreas insufficiency Age 0 to <25 years
Exclusion parameters	Acute liver failure Pre-transplantation status Choline supplementation via prescribed choline or dietary supplements 25 years or older No data on choline parameters

**Table 2 nutrients-17-00868-t002:** Biometric and clinical data of pancreas sufficient and insufficient CF patients.

Parameter	Exocrine Pancreatic Sufficiency	Exocrine Pancreatic Insufficiency
Number of patients (and data sets)	14 (31)	148 (446)
F508del compound heterozygous F508del homozygous Other	4 0 10	78 53 17
Median age at CF ∂iagnosis (y)	1.04 (0.92–5.34)	0.27 (0.10–1.67) ***p* = 0.0009**
Age at measurements (y)	13.03 (8.50–16.42) [0.89–21.90]	12.66 (7.22–16.36) [0.64–24.58] *p* = 0.8208
Sex (male/female)	8/6	68/80; *p* = 0.4258
Body weight (kg)	44.9 (28.8–56.8)	38.1 (22.2–54.6); *p* = 0.3634
Body length (cm)	156.0 (131.2–161.5) (71–181)	150.9 (117.0–165.0) (61–188) *p* = 0.7324
(BMI (kg/m^2^)	18.2 (17.2–21.4)	17.2 (15.8–20.1) *p* = 0.2199
BMI Percentile (%)	57.5 (26.8–73.3)	40.0 (18.6–64.8) *p* = 0.3051
Coagulation (Quick) [70–120]	95 (88–103)	89 (79–100) *p* = 0.0808
Thrombocytes (10^3^/µL)	281 (256–298)	304 (260–355) *p* = 0.2566
CRP (mg/dL) ESR (mm/h)	0.02 (0.01–0.04) ** 6 (4–8) **	0.02 (0.01–0.08) **; *p* = 0.7577 8 (5–14) **; *p* = 0.1572
Cholesterol (mg/dL) [130–190]	137(126–152)	128 (105–143) *p* = 0.3272
Triglycerides (mg/dL) [<200]	68 (55–74)	83 (62–117) *p* = 0.1218
Albumin (g/dL) [3.0–5.0]	4.3 (4.1–4.4)	4.0 (3.8–4.2) *p* = 0.0016
AST [<39 U/L]	19 (16–32)	25 (19–36) *p* = 0.0879
**ALT** [<39 U/L]	17 (13–20)	25 (20–34) ***p* = 0.0021**
AP [130–400]	225 (134–267)	225(150–279) *p* = 0.7037
gGT [<30 U/L]	12 (10–13)	12 (10–20) *p* = 0.2901
Lipid-soluble vitamins A (µmol/L) [1.1–2.7]	1.50 (1.15–1.70)	1.40 (1.20–1.60) *p* = 0.6669
E (µmol/L) [10–40]	24.3 (20.7–24.6)	20.4 (15.7–24.8) *p* = 0.0518
D (nmol/L) [50–175]	54.0 (46.6–59.0)	51.0 (35.0–63.8) *p* = 0.5444
Lung function parameters ppFEV1 (%) * FVC (%) * **FEF 25** (%) * FEF25–75 (%) *	97 (88–101) 103 (87–107) 104 (75–125) 93 (74–117)	96 (83–104) *p* = 0.9754 100 (93–107) *p* = 0.8155 80 (57–106) ***p* = 0.0450** 84 (64–100) *p* = 0.0843

Data are medians and interquartile ranges of 14 (pancreas sufficient) and 148 (pancreas insufficient) patients. Abbreviations: AST—serum aspartate aminotransferase; ALT—serum alanine aminotransferase; AP—alkaline phosphatase; CRP—C-reactive protein; ESR—erythrocyte sedimentation rate; gGT—gamma glutamyl transferase; ppFEV1—forced expiratory volume percent predicted; FVC—forced vital capacity; FEF25—forced expiratory flow at 25% of predicted FVC; FEF2575—forced expiratory flow at 25–75% of predicted FVC; *—median values of up to 6 independent measurements are indicated; **—as inflammation parameters were only determined in cases of suspected exacerbated infection, numbers are N = 10 and N = 9 for CRP and ESR in EPS patients, and N = 115 and N = 107 for CRP and ESR in EPI patients, respectively.

**Table 3 nutrients-17-00868-t003:** Choline compounds in plasma of study groups.

Parameter	EPS	EPI	*p*-Level
**A: Choline and water-soluble derivatives**			
Choline (µmol/L)	8.8 (8.0–10.0)	6.1 (5.2–7.4)	**<0.0001**
Betaine (µmol/L)	24.9 (21.4–27.1)	18.6 (14.8–24.6)	**0.0287**
Choline + Betaine	33.3 (30.9–35.6)	25.3 (20.5–31.9)	**0.0020**
TMAO (µmol/L)	1.4 (1.1–2.1)	4.9 (2.6–8.0)	**<0.0001**
Betaine/Choline	2.64 (2.36–3.31)	3.00 (2.473.66)	0.0772
TMAO/Choline	0.18 (0.13–0.20)	0.71 (0.44–1.35)	**<0.0001**
**B: Phospholipids**			
Phosphatidylcholine (PC) (mmol/L)	1.41 (1.33–1.50)	1.35 (1.19–1.51)	0.2927
Lyso-PC (% of PC)	2.57 (1.90–3.41)	2.56 (2.01–3.09)	0.7318
SPH (% of PC)	25.2 (21.1–29.3)	23.2 (20.6–26.2)	0.2344
Ceramides (% of PC)	0.27 (0.17–0.31)	0.24 (0.19–0.30)	0.6938
**C: PC sub-groups**			
Disaturated PC	1.53 (1.47–1.57)	1.70 (1.54–1.88)	**0.0056**
Oleyl (C18:1)-PC	17.4 (15.9–18.6)	18.9 (17.4–21.1)	**0.0031**
Linoleoyl (C18:2)-PC	50.5 (48.4–52.1)	47.9 (44.5–50.5)	**0.0121**
Arachidonoyl (C20:4)-PC	21.3 (19.2–23.8)	20.9 (18.9–23.0)	0.7543
Eicosapentaenoyl (C20:5)-PC	0.86 (0.79–0.98)	1.11 (0.88–1.37)	**0.0119**
Docosahexaenoyl: C22:6-PC	5.06 (4.46–6.69)	5.02 (4.34–6.29)	0.4949
Other PC *	2.55 (2.37–2.96)	2.43 (2.88–4.22)	**0.0005**

Data show medians and interquartile ranges of parameters in CF patients with exocrine pancreas sufficiency (EPS) (N = 14) compared to exocrine pancreas insufficient patients (EPI) (N = 148). Abbreviations: PC—phosphatidylcholine; SPH—sphingomyelin; TMAO—trimethylamine oxide; *, Other PC species include minor compounds, such as palmitoyl-palmitoleyl [C16:0/16:1]-, palmito-yl-linolenoyl [C16:0/18:3]-, stearoyl-docosatetraenoyl [18:0/22:4]- and stearoyl-docosapentaenoyl [C18:0/22.5]-PC.

## Data Availability

The original contributions presented in this study are included in the article/Appendix A. Further inquiries can be directed to the corresponding author.

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
