# Peer review of "Low Plasma Choline, High Trimethylamine Oxide, and Altered Phosphatidylcholine Subspecies Are Prevalent in Cystic Fibrosis Patients with Pancreatic Insufficiency"

_nutrients, 2025, doi:10.3390/nu17050868_

Round 1

Reviewer 1 Report

Comments and Suggestions for Authors

The introduction is fine. You could highlight the specifics on possible research gap regarding choline levels and bacterial overgrowth that this study aims to investigate.

Methodology is described well and in full detail. You should acknowledge in the discussion section the sample size imbalance.

Results are presented well. You could illustrate age-related trends in choline and TMAO levels.

In the discussion you should acknowledge the lack of direct SIBO measurements and suggest future research (e.g., microbiome sequencing). You could also discuss possible adverse outcomes for choline supplementation.

Author Response

The introduction is fine. You could highlight the specifics on possible research gap regarding choline levels and bacterial overgrowth that this study aims to investigate.

        Thank you! We have added this into Introduction (L. 99-101)

Methodology is described well and in full detail. You should acknowledge in the discussion section the sample size imbalance.

        Thank you! Our study was limited by imbalanced sample size, because 85-90% of CF patients are pancreas insufficient (EPI). Hence, in such a retrospective study on the routine patients in a CF outpatient clinic, imbalanced numbers of patients can only be eliminated by paired sample analysis. This, however, would have reduced the number of participants in the EPI group to 14 matched patients. We have addressed this in the Discussion (L. 336-340).

Results are presented well. You could illustrate age-related trends in choline and TMAO levels.

Thank you! The age-related trends are shown in Fig. 2A & C, showing the continuous decrease of choline, and no age-related trend for TMAO in EPI patients. The rho- and p-values for both parameters are provided in the figure.

In the discussion you should acknowledge the lack of direct SIBO measurements and suggest future research (e.g., microbiome sequencing). You could also discuss possible adverse outcomes for choline supplementation.

        Thank you for this comment. Increased TMAO levels and the known pathophysiological mechanisms suggest an altered gut microbiome in affected patients. However, as we did not routinely perform a culture-based measurement via the gold standard of quantification of colony-forming units of (colonic-type) bacterial flora present in a small bowel aspirate obtained by endoscopy, a correlation of TMAO levels and SIBO diagnosis remains speculative. However, as this may impact patients’ management, e.g. choline supplementation and/or SIBO therapy, future studies analyzing the gut microbiota in correlation to choline metabolism are needed. See corrected text L. 382-386.

        Adverse outcomes of choline supplementation in the dosages applied so far, i. e. below the upper limit according to IoM, are not described. However, this may not apply to patients without compliance to the formula and prescription, e.g. taking choline chloride outside from a meal and without adequate liquid. Notably, for adults the upper limit of choline is 3500g, i. e. 6.4fold the adequate intake of males (550mg/d) and 8.2fold the adequate intake of females (425mg/d) according to IoM/NAM. We have added safety and precaution informations as a separate para (4.5; L. 466ff)

Reviewer 2 Report

Comments and Suggestions for Authors

The manuscript is interesting, but I have the following comments:
I. Comments:
1. The title should be modified. It does not consider all the aspects evaluated in the patients.
2. Improve the wording of the objective of the study.
3. Introduction: 57 references in the introduction is too much. I suggest the authors rewrite the introduction, and leave 20 references as a maximum. Many of the references are redundant, and could be moved to the discussion.
4. The authors evaluated fatty acids and other metabolic parameters. Why did they not evaluate the subjects' diet?
5. The authors raise the importance of the microbiota. Why did they not do a study of the microbiota and analysis of short-chain fatty acids?
6. Could the results regarding PUFAs be explained by dietary intake, synthesis and/or deposits in the body?
7. Considering the main results, is it possible to establish a link with the inflammatory response?
7. What clinical intervention projection would this study have?

Author Response

Reviewer 2

The manuscript is interesting, but I have the following comments:

  1. Comments:
  2. The title should be modified. It does not consider all the aspects evaluated in the patients.

        We thank the reviewer for this evaluation. The title has been modified to “Low plasma choline, high trimethylamine oxide and altered phosphatidylcholine subspecies are prevalent in cystic fibrosis patients with pancreatic insufficiency”

  1. Improve the wording of the objective of the study.

        Thank you very much. The wording has been changed to “The goal of our study was to evaluate the plasma levels of choline, betaine, trimethylamine oxide (TMAO), PC, and PC sub-classes in CF patients from infancy to adulthood, and compare those with exocrine pancreatic insufficiency (EPI) to those with pancreatic sufficiency (EPS). (L. 18-21)

  1. Introduction: 57 references in the introduction is too much. I suggest the authors rewrite the introduction, and leave 20 references as a maximum. Many of the references are redundant, and could be moved to the discussion.

        Thank you very much for this comment. Indeed, 57 references are not necessary. However, eliminating all redundant literature ended up in 22 references. This is, to our opinion, the minimum of references to adequately address the background and rationale of our study.

  1. The authors evaluated fatty acids and other metabolic parameters. Why did they not evaluate the subjects' diet?

        Thank you for this comment! The main focus of this study was choline and its metabolites. We agree that evaluation of the patients’ diet is important for all essential nutrients (and has been addressed for additional choline via formula and other dietetic products). In this retrospective study, however, no validated protocols of food questionnaires on a routine basis were applied, and is not standard for choline and fatty acid profiles of all nutrition items. We discuss this in the limitations of study, and as a future step of investigation (Para 4.6, Limitations of study, L. 466ff).

  1. The authors raise the importance of the microbiota. Why did they not do a study of the microbiota and analysis of short-chain fatty acids?

Thank you for this comment. Indeed, analysis of the microbiota is important. However, the authors think that – similar to detailed food anamnesis on a continuous basis - this is a future step, based on the results of this study. We have included a para addressing the perspectives for future investigation (Para 4.5, L466ff) and limitations (4.6, L. 488ff)

  1. Could the results regarding PUFAs be explained by dietary intake, synthesis and/or deposits in the body?

        Thank you very much for this comment! Yes, this could well be, but again: this question is outside the main focus of the study, which is putative choline deficiency in patients. It is reasonable that differences in liver metabolism and nutrition between EPS and EPI patients may affect fatty acid metabolism. However, our study is the first at all to describe that such differences exist, and the data show that there is no difference in the concentrations/fractions of pro- and anti-inflammatory arachidonic and docosahexaenoic acid. We discuss it in L. 455-461.

  1. Considering the main results, is it possible to establish a link with the inflammatory response?

        This is an interesting point, thank you. However, inflammation markers, like C-reactive protein (CRP) and erythrocyte sedimentation rate (ESR) are not determined on a routine basis in our clinic, as they only have an impact on treatment in cases of suspected acute pulmonary infection. Calprotectin was not systematically determined either during the study period. However, we have analyzed available data on CRP and ESR for the limited number of patient samples. There was no significant differences between EPI and EPS patients, and correlations with choline, betaine, TMAO or phosphatidylcholine (PC) were not significant, with trends for choline and betaine in relation to ESR. We have included these data in the results (Table 2; L. 234-240), in the supplement (supplement figure S4 and S5) and in the discussion (L. 358-361).

  1. What clinical intervention projection would this study have?

        Thank you for this comment. The authors ‘believe’ that additional choline administration is preventive and curative for liver steatosis in CF patients, as previously shown in a pilot study and case report (ref. 18, 19). However, it will require randomized trials to prove this as a standardized routine treatment. While the authors are actually performing such a prospective randomized monocentric trial, we decide to include this ‘clinical intervention projection’ in the conclusions as an interventional option (see para 4.5, L. 466ff.)

Reviewer 3 Report

Comments and Suggestions for Authors

Major:

  1. The authors should strengthen the discussion on why clinical testing and choline supplementation should be routine. Are there existing clinical guidelines that support this approach, or would this study contribute to potential new recommendations?
  2. The exclusion of patients with additional choline intake is reasonable, but were any dietary records analyzed to confirm that patients were not receiving high amounts of choline from food sources?
  3. The study concludes that TMAO is increased early in life in CF patients with exocrine pancreatic insufficiency (EPI). Yet, no discussion is provided on potential age-specific factors influencing choline degradation. Could microbiota composition or delayed gut maturation contribute to these findings?
  4. The manuscript presents a detailed phosphatidylcholine (PC) subclass analysis, but how these changes translate into clinical outcomes is unclear. Does a lower linoleic acid PC fraction have any known impact on CF-related complications?

Minor:

  1. The figure legends should be more descriptive. For example, Figure 2D shows correlations, but the strength of the relationship is not mentioned.
  2. In some instances, the manuscript switches between abbreviations and full terms inconsistently (e.g., "TMAO" is sometimes referred to as "trimethylamine oxide").
  3. The manuscript frequently refers to unpublished data or internal references, which limits verification. Could the authors provide more external validation?
Comments on the Quality of English Language

Minor linguistic and stylistic corrections are required.

Author Response

Reviewer 3

Major:

    The authors should strengthen the discussion on why clinical testing and choline supplementation should be routine. Are there existing clinical guidelines that support this approach, or would this study contribute to potential new recommendations?

        The authors thank the reviewer for these suggestions. We have included this as an additional para (4.5-Perspectives …; L. 466ff)

    The exclusion of patients with additional choline intake is reasonable, but were any dietary records analyzed to confirm that patients were not receiving high amounts of choline from food sources?

        Thank you for this comment. No validated and systematic protocol using food questionnaires on a routine basis was applied that would allow for individual choline or fatty acid intake. Notably, choline and individual fatty acids are not a common routine part of dietary analysis, particularly not in CF patient care so far (Wilschanski et al. 2024. ESPEN-ESPGHAN-ECFS guideline on nutrition care for cystic fibrosis. Clin Nutr 43:413-445.). Moreover, phosphatidylcholine is the primary choline component in meat, fish and other nutrients except milk and supplements. They have different effects on absorption kinetics, TMAO formation etc. (see ref 44). Patients were asked, whether they took any supplements. We discuss this as a future step of investigation and in the limitations of study (Para 4.5, l. 466ff; para 4.6, l. 487ff). See also answer to comment 4, reviewer 2)

    The study concludes that TMAO is increased early in life in CF patients with exocrine pancreatic insufficiency (EPI). Yet, no discussion is provided on potential age-specific factors influencing choline degradation. Could microbiota composition or delayed gut maturation contribute to these findings?

        Thank you for this comment. Indeed, the intestinal microbiota is an important aspect of intraluminal choline degradation. Again, in this retrospective study we did not include all parameters, and microbiota analysis in relation to choline deficiency surely is a future project. We have included this in the discussion (L. 422-426).

    The manuscript presents a detailed phosphatidylcholine (PC) subclass analysis, but how these changes translate into clinical outcomes is unclear. Does a lower linoleic acid PC fraction have any known impact on CF-related complications?

        Thank you for this point! Indeed, we only describe our findings, with no obvious clinical impact. This applies for the consistency in PC containing arachidonic or docosahexaenoic acid that remain unchanged in spite of linoleic acid and EPA changes. See L. 277f.

Minor:

    The figure legends should be more descriptive. For example, Figure 2D shows correlations, but the strength of the relationship is not mentioned.

        Thank you, we changed the figure legends. Correlation coefficients and p-values are indicated in all parts of Fig. 2D. Additionally, in the text we point out that in spite of its significance level the correlation is only weak (rho= = -0.2666; L. 266f).

    In some instances, the manuscript switches between abbreviations and full terms inconsistently (e.g., "TMAO" is sometimes referred to as "trimethylamine oxide").

        Has been corrected. (L. 237, 250).

    The manuscript frequently refers to unpublished data or internal references, which limits verification. Could the authors provide more external validation?

        We have checked the manuscript and find no reference to unpublished data in the revised version of the manuscript.

Round 2

Reviewer 2 Report

Comments and Suggestions for Authors

The authors responded to all my comments and suggestions. Therefore, the manuscript can be accepted. I have only one final comment. The authors could include a figure (in the discussion) summarizing the main results obtained. I believe that this would improve the understanding of the manuscript.

Reviewer 3 Report

Comments and Suggestions for Authors

The authors have thoroughly addressed all my comments.